# Effect of Mulberry Leaf Powder of Varying Levels on Growth Performance, Immuno-Antioxidant Status, Meat Quality and Intestinal Health in Finishing Pigs

**DOI:** 10.3390/antiox11112243

**Published:** 2022-11-14

**Authors:** Jiayu Ma, Hong Ma, Sujie Liu, Jian Wang, Hongliang Wang, Jianjun Zang, Shenfei Long, Xiangshu Piao

**Affiliations:** 1State Key Laboratory of Animal Nutrition, College of Animal Science and Technology, China Agricultural University, Beijing 100193, China; 2Ministry of Agriculture Key Laboratory of Animal Virology, Department of Veterinary Medicine and Center of Veterinary Medical Sciences, Zhejiang University, Hangzhou 310000, China; 3College of Resources and Environmental Sciences, China Agricultural University, Beijing 100193, China

**Keywords:** mulberry leaf powder, anti-oxidative properties, meat quality, microbiota, finishing pigs

## Abstract

Background: The purpose of the research was to investigate the effect of mulberry leaf powder (MP) of varying levels on growth performance, immuno-antioxidant status, meat quality and intestinal health. A total of 120 healthy finishing pigs (Duroc × [Landrace × Yorkshire], 72.2 ± 4.8 kg) were selected for this experiment and divided into four treatments, according to randomized complete block design, with three replicates of 10 pigs each (barrow:gilt = 1:1). The dietary treatments were as follows: (1) Ctrl, corn-soybean meal basal diet; (2) MP_6: basal diet with 6% MP. (3) MP_9: basal diet with 9% MP; (4) MP_12, basal diet with 12% MP. The whole experiment lasted for 33 days. Results: Compared to the Ctrl, dietary supplementation of 9% MP had no negative effects on growth performance, antioxidative capacity, intestinal digestive enzyme activities and inflammatory factors, carcass trait, the lipid profile and amino acid of muscle or related mRNA expression. MP_6 and MP_12 showed detrimental effects on average daily gain (*p <* 0.05) and digestibility (*p <* 0.05) of dry matter, crude protein and gross energy. Furthermore, MP_9 could improve (*p <* 0.05) the level of serum immunoglobulin M, enhance (*p <* 0.05) the a*_24h_ of meat, up-regulate (*p <* 0.05) the mRNA expression of occludin and Mucin-2, increase (*p <* 0.05) the relative abundance of *Bifidobacterium* and decrease (*p <* 0.05) the relative abundance of *Campylobacter* in the hindgut. Conclusion: Summarizing our study, dietary supplementation of 9% MP had no negative effects on growth performance, antioxidative capacity, intestinal digestive enzyme activities and inflammatory factors, carcass trait, the lipid profile and amino acid of muscle, or related mRNA expression. Furthermore, MP_9 could improve serum immunity, enhance meat quality, up-regulate the mRNA expression related to the mechanical and chemical barriers and enrich the beneficial microbiota of the hindgut. Hence, dietary supplementation of 9% MP in finishing pigs may be advisable.

## 1. Introduction

With the increasing demand for livestock products, the feedstuffs are being tightened. Accordingly, numerous cost-effective alternatives concerning feed sources are emerging for the sustainable development of the livestock industry. Many researchers have demonstrated in recent years that unconventional feed ingredients, such as brown rice, corn germ meal, wheat gluten meal and distillers dried grains with solubles, are available as partial substitutes for grain-based concentrates in livestock feeds without adverse effects on animal growth performance [1,2]. Moreover, there are multiple benefits of unconventional feed ingredients, with the most notable being their antimicrobial and antioxidant activities [3].

Mulberry trees (*Morus alba* L.) are rapid-growing, long-living, mow-resistant and deciduous plants with high adaptability to climate and soil, which can be used for 20–30 years after planting once and can be mowed 3–4 times annually. The cultivated area of mulberry in China is approximately 106 hectares [4], and the annual fresh mulberry leaves weight can reach 80–120 tons/hectares in areas with fertile and well-irrigated soil. The principal active components of mulberry leaves, such as flavonoids and polyphenols, have anti-inflammatory, antioxidant, anti-diabetic, hypolipidemic and neuroprotective properties [5,6], which have been utilized in Chinese herbal medicine for centuries, such as for the treatment of cough, fever, diabetes and rheumatic diseases [7,8]. Research on the use of mulberry leaves for feed was initiated abroad as early as the beginning of the 20th century, and the Food and Agriculture Organization of the United Nations (FAO) has given high priority to the utilization of mulberry resources for the development of livestock [9]. Related studies have indicated that mulberry leaf powder (MP) can be supplemented as a potential protein source for cattle [10] and as a fermentable energy and potential protein source for sheep [2,11]. The findings of a study concerning rumen and gastrointestinal digestibility in sheep demonstrated that the digestible energy and crude protein values of mulberry leaves were comparable to those of alfalfa hay [12]. Furthermore, mulberry leaves can be used as a new feed supplement to modulate the antioxidant capacity of laying hens [13]. China promulgated and implemented “Mulberry leaf powder for feed” (SB/T 0998-2013) in 2013, aiming to actively develop mulberry resources and expand its utilization pathways. Nevertheless, the effect of MP on the immuno-antioxidant properties and meat quality of finishing pigs has been seldom investigated, especially for intestinal health.

In the current study, we aim to evaluate the effect of MP of varying levels on growth performance, serum biochemistry and immuno-antioxidant characteristics, meat antioxidant and lipid profile, with a finishing pig model, especially for intestinal morphology, tight junction expression and microbiotas, which provided a basis for future utilization and optimization of MP in livestock.

## 2. Materials and Methods

All procedures of the present research were approved by the Institutional Animal Care and Use Committee of China Agricultural University (No. AW10601202–1-2, Beijing, China).

### 2.1. MP Product

The MP was supplied by the Manborui Biotechnology Co., Ltd. (Jingzhou, Hubei, China), which is primarily obtained from the leaves, buds and some young branches of mulberry trees. The mulberry leaves and shoots with a harvested length of 45–60 cm were laid directly in a dry and ventilated place, shade-dried to a moisture content of approximately 30%, then spread out in the sun or into a drying room to dry quickly, lowering the moisture to 8–12%, and crushed using a feed or straw grinder. The nutritional composition of MP is shown in Appendix A.

### 2.2. Experimental Design, Animals and Management

A total of 120 healthy finishing pigs (Duroc × [Landrace × Yorkshire]; 72.2 ± 4.8 kg) were selected for this experiment and divided into 4 treatments, according to randomized complete block design with 3 replicates of 10 pigs each (barrow:gilt = 1:1). The dietary treatments were as follows: (1) Ctrl, corn-soybean meal basal diet; (2) MP_6: basal diet with 6% MP. (3) MP_9: basal diet with 9% MP; (4) MP_12, basal diet with 12%MP. The whole experiment lasted for 33 days and the formulations of experimental diets were shown in Appendix A, which either satisfied or excelled the NRC [14] of finishing pigs.

The research was conducted at the Fengning animal experimental base of China Agricultural University (Chengde, China). The experiment was carried out in a fully enclosed finishing house, which was thoroughly cleaned and disinfected before the experiment and managed with an all-in/all-out feeding pattern. The finishing house was equipped with intelligent equipment to control the temperature (20 °C), humidity (75%) and ventilation intensity, and the house floor was semi-cement and semi-leakage. The chosen pigs were randomly allocated to 12 pens (2.7 m × 1.8 m × 0.9 m) with the same equipment and conditions per pen, all of which used stainless steel adjustable troughs and nipple-type waterers. The finishing pigs were fed and drank *ad libitum* during the experiment, and the health status of the finishing pigs was observed daily. Moreover, routine deworming and immunization were necessary, according to the management procedures of the base. The whole experiment was performed in strict accordance with the standards of animal welfare of China Agricultural University.

### 2.3. Growth Performance

The amounts of feed intake and residual feed for finishing pigs were logged daily on a pen basis to calculate the average daily feed intake (ADFI) and the selected pigs were weighed on D 0, D 20 and D 33 to calculate the average daily gain (ADG), then the feed conversion ratios were calculated (FCR = ADFI/ADG).

### 2.4. Sampling and Slaughtering

#### 2.4.1. Feed and Fecal Samples

At the beginning, middle and end of the experiment, approximately 1 kg of feed samples were collected for each treatment, then mixed well and stored at −20 °C for analysis.

On the 30th day of the experiment, manure was completely cleaned in the fattening house. From the 31st to the 33rd day, fecal samples (approximately 1 kg) were collected in each replicate, and the contamination should be avoided during the collection process. Finally, the collected fecal samples were well-mixed and stored at −20 °C for further analysis.

#### 2.4.2. Serum Sample

The blood (10 mL in the anterior vena cava) was collected from one finishing pig with near average body weight in each replicate on the morning of the 20th and 33rd days of the experiment. The blood sample was centrifuged at 3000× *g* for 10 min at 4 °C after standing for 3 h, then collected the serum and stored at −20 °C for examination.

#### 2.4.3. Slaughtering

At the end of the feeding experiment, one pig with near average body weight was picked from each pen, transported to the slaughterhouse (about 1 h) on the next morning after 12 h of fasting and rested for 4 h, then bled to slaughter by electric shock.

After slaughter, the longissimus dorsi muscle of the left carcass and liver were gathered; the intestinal mucous membrane samples from the duodenum, jejunum and ileum at 1/3 of the posterior segment in finishing pigs were washed off with 0.9% saline gently and collected, respectively, then stored at −80 °C. Meanwhile, approximately 2 cm tissues from duodenum, jejunum and ileum at the middle of the segment were collected, respectively, and fixed in 4% paraformaldehyde for examining morphology. Additionally, approximately 10 mL of the cecum and colon digesta were collected, respectively, transferred to the liquid nitrogen tank and later stored at −80 °C. All samples were obtained in duplicate.

### 2.5. Feed/Fecal Nutrient Composition

The fecal samples were defrosted at 4 °C and subsequently dried at 65 °C within an oven for 72 h. The feed and fecal samples were crushed and sieved through 40 mesh. Dry matter (DM), ash, crude protein (CP) ether extract (EE), calcium (Ca) and phosphorus (P) were determined based on the methods of the Association of Official Analytical Chemists (AOAC) [15], respectively. Organic matter (OM) was calculated (OM (%) = 1-Ash (DM-basis) × 100%). Neutral detergent fiber (NDF) and acid detergent fiber (ADF) were determined with reference to the method of van Soest et al. [16] using a filter bag and ANKOM200 fiber analyzer (Ankom, USA). Gross energy (GE) was analyzed using a 6400 automatic isoperibol calorimeter (Parr, USA). Chromium (Cr) level was determined by Z-5000 atomic absorption spectrophotometer (Hitachi, Japan), according to the methodology of Williams et al. [17], to calculate the apparent total tract digestibility (ATTD) of nutrients with the following equation:ATTD (%) = 1 − (Cr _feed_ × Nutrient _feces_)/(Cr _feces_ × Nutrient _feed_)

### 2.6. Serological Analysis and Intestinal Enzyme Activity

Serum immunoglobulins (IgA, IgG, IgM), inflammatory factors interleukin (IL-1β, IL-6, IL-8, IL- 10), tumor necrosis factor-α (TNF-α), gamma-interferon (IFN-γ), D-lactate (D-LA), hormones, glucocorticoid (GC), adrenocorticotropic hormone (ACTH), epinephrine (EPI), epidermal growth factor (EGF) and non-esterified fatty acid (NEFA), as well as intestinal inflammatory factors (IL-1β, IL-4, IL-6, IL-10 and TNF-α) and secretory immunoglobulin A (sIgA), were analyzed using Multiskan Ascent fully automated enzyme marker (Thermo, Waltham, USA) with commercial ELISA test kits. Serum growth hormone, insulin and insulin-like growth factor-1 (IGF-1) levels were determined by the methodology of radioimmunoassay using a DFM-96 radioimmunoassay gamma counter (Zhongcheng Electromechanical Technology, China). The glucose (GLU), total cholesterol (TC), total triglycerides (TG), high-density lipoprotein (HDL), low-density lipoprotein (LDL), alanine aminotransferase (ALT), aspartate aminotransferase (AST), total protein (TP), albumin (ALB), globulin (GLB), alkaline phosphatase (AKP), lactate dehydrogenase (LDH), uric acid (UA), blood urea nitrogen (BUN) in serum and the superoxide dismutase (SOD), glutathione peroxidase (GSH-Px), total antioxidant capacity (T-AOC), catalase (CAT) and malondialdehyde (MDA) in serum and liver were analyzed by a CLS880 fully automatic biochemical analyzer (Zecen Biotech, China). The determination of amylase, lipase, trypsin and chymotrypsin activities in the pancreas, jejunum and ileum of finishing pigs was performed using immunoturbidimetric assay.

All commercial kits were acquired from Nanjing Jiancheng Institute of Biological Engineering (Nanjing, China) and all analytical procedures were followed in strict compliance with the manufacturer’s instructions.

### 2.7. Determination of Carcass Trait and Meat Quality

After slaughtering the pigs, the head, hooves, tail and offal were completely removed and the hot carcass weight was recorded to calculate the slaughter rate. The back fat thickness was measured using a vernier caliper at shoulder fat thickness, the last rib fat thickness, lumbosacral fat thickness, the 6th to 7th rib fat thickness and the 10th rib fat thickness, respectively. Furthermore, the length and width of the longissimus dorsi cross-section were measured to calculate the loin-eye area. The pH values of the longissimus dorsi were recorded at 45 min and 24 h using a 205 pH meter (Testo, Lenzkirch, Germany). The flesh color luminosity (L*), redness (a*) and yellowness (b*) of the samples was determined using a CR-410 colorimeter (Minolta, Osaka, Japan). The longissimus dorsi was scored using official color and marbling standards card (NPPC, Des Moines, IA, USA). Drip loss is the percentage of water reduction after 100 g of meat samples were suspended in a refrigerator at 4 °C for 24 h. The relevant calculation formula is as follows:Loin-eye area (cm^2^) = LEL × LEW × 0.7
Dressing percentage (%) = HW/LW × 100%
Drip loss (%) = (Mm − Mh)/Mm × 100%

Note: LEH, loin-eye length; LEW, loin-eye width; HW, hot carcass weight; LW, live weight; Mm, meat weight; Mh, meat weight after heating.

### 2.8. Muscle Lipid Profile and Amino Acid

The fatty acid and amino acid concentration assays of meat samples were performed with the assistance of the Ministry of Agricultural Feed Industry Center of China Agricultural University. The meat samples need to be lyophilized for 60 h by lyophilizer and thawed first, then quantification can take place in accordance with the methods of Sukhija and Palmquist [18]. The detailed procedures were as follows: 0.5 g of the sample was weighed accurately into a hydrolysis tube; supplemented with 4 mL of 10% chloroacetylmethanol solution and 1 mL of undecanoic acid, (1.0 mg/mL) as an internal standard solution; then 1 mL of n-hexane was added and capped tightly; and put into a thermostat water bath at 80 °C for 2 h. After cooling, the sample was supplemented with 7% potassium carbonate 5 mL and shaken evenly and centrifuged at 1000 r/min for 5 min, then the lipid fatty acid of the muscle was determined using 6890 N gas chromatography (Agilent, Santa Clara, CA, USA) after the samples passed through the 0.2 μm filter membrane.

The determination of amino acids in muscle was performed by hydrochloric acid hydrolysis method. Firstly, 0.1 g of dried muscle sample powder was accurately weighed into an ampoule, 10 mL of 6 mol/L hydrochloric acid solution was added, the ampoule on the flame of an alcohol lamp was sealed, then it was hydrolyzed in a thermostat water bath at 110 °C for 24 h. After cooling and mixing, it was filtered and transferred to a 100 mL volumetric flask, then 1 mL of liquid was pipetted into a 10 mL centrifuge tube and evaporated in a thermostat water bath at 60 °C, before being supplemented with 1 L of distilled water to dissolve the residue, and it then continued to evaporate, being repeated twice to remove the hydrochloric acid and repeated twice to remove the hydrochloric acid completely. Later, 1 mL of double-distilled water was added and shaken evenly, the content of amino acid was determined using L-8900 fully automatic amino acid analyzer (Hitachi, Tokyo, Japan) after the samples were filtered through a 0.45 m microporous membrane.

### 2.9. Intestinal Morphometry

After being fixed in 10% formaldehyde solution for 48 h, sections (5 μm) were washed, excised, and dehydrated in ethanol, transparent in xylene and embedded in paraffin, and the sections were stained with hematoxylin-eosin; the images of the sections were acquired using CX31 light microscopy (Olympus, Tokyo, Japan) combined with true color image analysis software; the multiple typical fields of view (with clear and intact villi) were chosen on each section to measure villi height (VH) and crypt depth (CD), and the ratio of villi height to crypt depth (VH/CD) was calculated.

### 2.10. RNA Extraction and Real-Time PCR

Total RNA was extracted, respectively, from the liver, muscle, jejunum and ileum of finishing pigs by TRizol (Invitrogen, Carlsbad, USA) kits, as described in the manufacturing instructions and the concentration and quality of RNA were verified by 1% agarose gel electrophoresis. The RNA was reverse transcribed into cDNA with All-in-One First-Strand cDNA Synthesis SuperMix for qPCR kit (QIAGEN, Hilden, Germany). Reverse transcription system: total RNA, 0.5 μg; 5 × *TransScript^®^* All-in-one SuperMix for qPCR, 5 μL; gDNA Remover, 0.5 μL; RNase-free water supplemented to 10 μL. Reaction system: 42 °C, 15 min; 85 °C, 5 s. An amount of 90 μL RNase-free water was added after reverse transcription and stored at −20 °C.

Fluorescent quantitative PCR procedures were conducted using the Roche LightCycler^®^ 480II Real-Time PCR System (Roche, Basel, Switzerland; PCR efficiency: 96~l02%). PCR reaction mixture (10 μL): cDNA, 1 μL; 2× PerfectStart^TM^Green qPCR SuperMix, 5 μL; upstream primer, 0.2 μL; downstream primer, 0.2 μL; RNase-free water, 3.6 μL. Then, it was incubated using a Real-Time PCR 384-well plate (Roche, Basel, Switzerland). Reaction system: pre-denaturation at 95 °C for 5 min; denaturation at 95 °C for 10 s, renaturation at 60 °C for 30 s; extension at 72 °C for 30 s, 40 cycles. The melting curves were analyzed at the end of the PCR cycle to validate the specific generation of the expected PCR products. Each sample was analyzed twice. The primer sequences listed in Appendix A were designed and synthesized by Beijing Tianyi Huiyuan Biotechnology (Beijing, China). The expression levels of mRNA were standardized using β-actin (housekeeping gene) and the relative expression of mRNA was calculated using the 2^−ΔΔCt^ method.

### 2.11. The 16S rRNA Microbial Sequencing

The 16S rRNA microbial sequencing was accomplished with the collaboration of Shanghai Majorbio (China). The samples of the cecum and colon digesta were taken from the −80 °C refrigerator, and the DNA kit (OmegaBio-Tek, Norcross, GA, USA) was used for extracting the total genomic DNA of bacteria. The concentration and purity of DNA were examined by NanoDrop2000 spectrophotometers (Thermo, Waltham, MA, USA). The 16S rRNA gene in the variable region of bacterial V3-V4 was amplified with PCR, in accordance with universal primers 338F (5-ACTCCTACGGGAGGCAGCAG-3) and 806R (5-GGACTACHVGGGTWTCTAAT-3). Then, the amplification products were purified and recovered by Axyprep DNA Gel Extraction kit (Axygen Biosciences, Union, CA, USA) and quantified to homogeneous concentrations by Qubit2.0 fluorometer (Thermo, Waltham, MA, USA). The purified amplified fragments were established as amplified libraries and sequenced on Illumina-HiSeq-PE300 platform (Illumina, San Diego, CA, USA) with paired-end reads of 300 bp. The raw sequences were quality-controlled by Fastp, spliced by FLASH [19], operational taxonomic unit (OTU) clustered by UPARSE based on the UPARSE algorithm with a 97% similarity threshold [20,21], and Taxonomic analysis by RDP classifier based on Bayesian algorithm. Lastly, the identified classifications were aligned using the Silva 16S rRNA database with a 70% confidence threshold [22]. The Sobs, Shannon, Simpson, Ace, Chao and phylogenetic diversity (PD) indices were analyzed by Mothur to assess microbial α-diversity. Principal co-ordinate analysis (PCoA) and partial least squares discriminant analysis (PLS-DA) based on the Bray–Curtis distance matrix algorithm and analysis of similarity (ANOSIM) were analyzed by R software to evaluate microbial β-diversity; Circos diagrams were plotted by the mapping software Circos to reveal the proportion of dominant species composition in each group and the proportion of distribution of each dominant species in different groups. The relative abundance of microbes was presented as a percentage. The relevant software information is listed below: Fastp (v0.23.2, https://github.com/OpenGene/fastp, accessed on 7 April 2022); Flash (v1.2.11, https://ccb.jhu.edu/software/FLASH/index.shtml, accessed on 7 April 2022); UPARSE (v 7.1, http://drive5.com/uparse/, accessed on 7 April 2022); RDP classifier (v 2.13, http://rdp.cme.msu.edu/, accessed on 7 April 2022); Silva 16S rRNA database (v138, http://www.arb-silva.de, accessed on 7 April 2022); Mothur (v1.30.2, https://mothur.org/wiki/calculators/, accessed on 27 April 2022); Circos (v 0.69.9, http://circos.ca/ accessed on 21 May 2022). Lefse analysis based on the nonparametric factorial Kruskal–Wallis sum test and Wilcoxon rank sum test with all-against-all multi-group comparison strategy was used to estimate features with significant differences in abundance and to identify taxa with significantly different abundances and to show only taxa with LDA scores larger than 3. The raw microbial dataset has been uploaded on NCBI with the accession number PRJNA890454 and PRJNA890458.

### 2.12. Statistical Analysis

All data were calculated as individuals, except for growth performance data, which were calculated based on pens. Source data were initially organized by Excel (Microsoft, Redmond, WA, USA). The one-way ANOVA with generalized linear models (GLM) procedure and the Turkey–Kramer post hoc test were performed by SAS 9.2 software (SAS Institute, Cary, NC, USA). The linear and quadratic comparisons were employed to establish the dose effect of MP in finishing pigs. Consideration was assigned to statistical significance when *p* ≤ 0.05 and a trend with statistical significance was regarded when 0.05 < *p* ≤ 0.10.

## 3. Results

### 3.1. Growth Performance

As presented in Table 1**,** From D 1 to D 20 and D 21 to D 33, there were slight declines in ADG and ADFI of finishing pigs between the treatments and the Ctrl, but no significant differences were noted. From D 1 to D 33, MP_6 and MP_12 decreased (*p* < 0.05) the ADG of finishing pigs, and for MP_9, no difference was observed.

### 3.2. Serum Immunity, Inflammatory Factors and Metabolites

As presented in Table 2 and Table 3, on D 22, No significant change in serum immunity of finishing pigs dietary MP supplementation; for inflammatory factor, MP_12 enhanced (*p* < 0.05) the level of IFN-γ. For serum metabolite, the level of GLU was linearly declined (*p* < 0.05) in MP_6, MP_9 and MP_12. Additionally, MP_12 enhanced (*p* < 0.05) the content of HDL and reduced the level of ALP.

On D 33, MP_9 increased (*p* < 0.05) the level of IgM, but there was no differences on the inflammatory factors between treatments and Ctrl. For serum metabolite, MP_9 decreased (*p* < 0.05) the level of GLB compared with the MP_6, but no difference compared with the Ctrl.

### 3.3. Antioxidant Characteristics and Serum Hormone Contents

As presented in Figure 1**,** dietary MP supplementation had no effect on antioxidant capacity whether serum or liver of finishing pigs. Nevertheless, a higher level (*p* < 0.05) of T-AOC, as well as a lower (*p* < 0.05) level of MDA, was noticed in MP_9, compared with MP_6. On D 20, an increased (*p* < 0.05) level of GC and ACTH and were noticed in MP_9, compared with MP_6, and MP_9 increased the level of IGF-1, compared with the MP_6 and MP_12. Moreover, MP_12 improved (*p* < 0.05) the level of GC, compared with MP_6. Nonetheless, no difference was noted when compared with the Ctrl. On D 33, an increased level of ACTH was observed compared with the Ctrl and MP_6.

### 3.4. Nutrient Digestibility

As presented in Table 4, MP_6 and MP_12 lowered (*p* < 0.05) the ATTD of DM, CP and GE. Nevertheless, with regard to nutrient digestibility in finishing pigs, no significant difference was observed between MP_9 and the Ctrl.

### 3.5. Carcass Characteristics and Meat Quality

As presented in Table 5, no differences were noticed between treatments and the Ctrl on carcass characteristics. For meat quality, MP_9 improved (*p* < 0.05) the a*_24h_ of meat compared to MP_12, but there was no difference compared to the Ctrl.

### 3.6. Muscle Amino Acid and Lipid Profile

As presented in Table 6 and Table 7, no differences were observed between treatments and the Ctrl on muscle fatty acid and lipid profile in finishing pigs, indicating dietary MP supplementation might have no effect on the concentration of muscle amino acid and lipid profile of finishing pigs.

### 3.7. Intestinal Digestive Enzyme Activity, Inflammatory Factors and Morphological Analysis

As presented in Figure 2, no differences were noted on digestive enzyme activity and inflammatory factors of finishing pigs dietary MP supplementation. However, the trypsin and lipase activities dramatically declined (*p* < 0.05) in MP_12, compared to the MP_9.

As presented in Table 8, for intestinal morphology, the duodenal and jejunal villus height were reduced with increasing doses of MP, but there was no significant difference between treatments and the Ctrl.

### 3.8. Relative mRNA Expression Associated with Intestinal Function and Antioxidant Capacity, As Well As Muscle Fiber and Lipid Metabolism

As presented in Figure 3, for the jejunum of finishing pigs, a higher (*p* < 0.05) relative mRNA expression of tight junction proteins occludin and Mucin-2 were noticed. Nonetheless, dietary MP supplementation in finishing pigs shows neither a clear effect on the mRNA expression of antioxidant enzyme-related genes in the liver and muscle nor the muscle fiber and lipid metabolism-related genes.

### 3.9. Cecal and Colonic Microbial Sequencing and α-Diversity

For cecal microbiota, twelve cecal digesta samples of finishing pigs were sequenced and analyzed, and a total of 821,704 optimized sequences with 415 bp average length were acquired. After a random flattening with the minimum sample sequence, a total of 924 OTUs were observed and classified into 16 phylum, 25 classes, 47 orders, 78 families, 198 genus and 377 species, with the similarity compared with the Silva database. Similarly, for colon microbiota, a total of 887,748 optimized sequences with 414 bp average length were acquired and 1,106 OTUs were recognized and classified into 17 phylum, 27 classes, 52 orders, 90 families, 214 genera, and 433 species.

Nevertheless, as presented in Figure 4, no significant differences were identified between the treatments and the Ctrl on α-diversity indices in the cecum and colon of finishing pigs.

### 3.10. Cecal and Colonic Microbial Composition and β-Diversity Analysis

As presented in Figure 5, for the cecum of finishing pigs, there were 573 common OTUs, and 36 (Ctrl), 19 (MP_6), 19 (MP_9) and 6 (MP_12) unique OTUs, respectively, from Venn analysis (Figure 5A). The microbial composition was visualized by barplot and heatmap at the phylum (Figure 5C), family (Figure 5D) and genus levels (Figure 5E). The top five microbiota at the phylum level in the Ctrl were *Firmicutes* (62.87%), *Bacteroidota* (29.93%), *Proteobacteria* (2.61%), *Campilobacterota* (2.23%) and *Spirochaetota* (1.28%). The top five microbiota at the family level in the Ctrl were *Prevotellaceae* (22.94%), *Clostridiaceae* (7.35%), *Lachnospiraceae* (12.57%), *Oscillospiraceae* (11.22%) and *Peptostreptococcaceae* (5.07%). The circos diagram at the family level (Figure 6A,D) was plotted to explicitly present the proportion of dominant species distribution in each treatment, together with the proportion of each dominant species distribution in different treatments. At the genus level, the *Clostridium_sensu_stricto_1* (7.16%), *UCG-005* (6.64%), *Alloprevotella* (6.58%), *Prevotella* (6.06%) and *Terrisporobacter* (4.79%) were dominated. There were no significant differences in microbial structure between the treatments and Ctrl at the family and genus levels, which were also proved by the PCOA (*r* = 0.1049, *p* = 0.231) and PLS-DA results (Figure 7B,C) on the basis of the bray_curtis algorithm and ANOSIM difference examination.

Furthermore, the LEfse analysis (Figure 7A) indicated that the cecum of finishing pigs in the Ctrl were enriched in *Campylobacter* and *Negativibacillus*, and MP_12 were enhanced in *Senegalimassilia*. According to the results of the Kruskal–Wallis H test with falsely discovery rate (FDR) correction and Tukey–Kramer post hoc test, the relative abundance of *Campilobacterota* (Figure 7C), *Campylobacteraceae* (Figure 7D), *Campylobacter* and *Negativibacillus* (Figure 7E) was lower (*p* < 0.05) in the cecum of finishing pigs supplemented MP and the relative abundance of *Senegalimassilia* in MP_12 was increased (*p* < 0.05).

For the colon of finishing pigs, there were 671 common OTUs and 58 (Ctrl), 10 (MP_6), 20 (MP_9) and 9 (MP_12) unique OTUs, respectively, from Venn analysis (Figure 5B). The dominated microbiota at the phylum level (Figure 5F) in the Ctrl were *Firmicutes* (73.33%), *Bacteroidota* (23.00%) and *Spirochaetota* (1.25%). The primary microbiota at the family level (Figure 5G) in the Ctrl were *Prevotellaceae* (14.53%), *Oscillospiraceae* (13.77%), *Clostridiaceae* (10.09%), *Lachnospiraceae* (9.18%), *Lactobacillaceae* (5.95%) and *Peptostreptococcaceae* (4.45%). At the genus level (Figure 5H), the *Clostridium_sensu_stricto_1* (9.93%), *Lactobacillus* (5.95%), *Prevotella* (5.73%), *norank_f__Muribaculaceae* (4.45%) and *Terrisporobacter* (4.22%) were dominated. In addition, the results of PCOA (*r* = 0.1636, *p* = 0.130) and PLS-DA (Figure 6E,F) indicated that dietary MP supplementation had no effect on the microbial microbiota of colons in finishing pigs. The results of LEfse analysis (Figure 7B) presented that the colon of finishing pigs in the Ctrl were enriched in *Lactobacillus*; MP_9 were enhanced in *Bifidobacterium* and MP_12 were enriched in *Hydrogenoanaerobacterium*, *Colidextribacter* and *Catenibacterium.* Based on the results of Kruskal–Wallis H test, the relative abundance of *Lactobacillaceae* (Figure 7F) and *Lactobacillus* (Figure 7G) was lower (*p* < 0.05) in MP_6 and MP_12. However, the difference between MP_9 and the Ctrl was not significant.

## 4. Discussion

Mulberry leaves are wealthy in carbohydrates, proteins, fatty acids and fiber. Additionally, the vitamins and mineral elements, especially amino acids are abundant, as well as containing numerous natural active substances, such as sitosterol, quercetin, γ-aminobutyric acid, 1-deoxynojirimycin and mulberry flavonoids [23], suggesting that mulberry leaves may be suitable as functional feed to enhance performance and health of finishing pigs. Recently, the impact of the epidemic has led to continuously elevated prices of traditional feed ingredients, which has brought concern for many businesses. Accordingly, the application of mulberry leaves as an unconventional feed source for livestock to substitute the traditional expensive feed ingredients (alfalfa, peanut meal, wheat bran) in animal nutrition is gradually gaining attention. In the present study, diets supplemented with 9% MP had no difference on ADG, ADFI and FCR, indicating that dietary supplementation of MP with 9% may have had no negative effect on the growth performance of finishing pigs, which was in agreement with the findings on pigs and cattle by Liu et al. [24] and Vu et al. [10]. However, diets supplemented with 6% and 12%, the ADG of finishing pigs decreased, which was in line with Zhu et al. [25] who suggested that diets supplemented with 15% MP decreased the ADG of finishing pigs in the whole period. The reason may be attributed to the lower amount of essential amino acids (6%) and the high concentration of fiber (12%) in the MP, compared with wheat bran [26]; the growth-promoting effect of MP_9 is most likely correlated with the positive effect of the increased active substances, such as flavonoids and polyphenolic compounds although this requires further validation. Research has revealed that high-fiber diets have a detrimental effect on energy and nutrient absorption [27], leading to decreased growth performance in finishing pigs [28]. Hence, it is necessary to consider lowering the fiber level or enhancing the essential amino acid level in MP to obtain a superior growth performance in finishing pigs.

To further investigate the effect of the dietary supplementation of MP on the growth in finishing pigs, the blood was collected at days 20 and 33 of the experiment and analyzed for immuno-antioxidant capacity and biochemical metabolic indices, as well as hormone levels. Interestingly, in the current study, an increased level of IFN-γ, as well as the decreased content of ALP in MP_12. ALP improves bone-portion and accelerates the deposition of Ca and P in bone tissue [29]. IFN-γ, as a pro-inflammatory cytokine, is capable of promoting its inflammatory response and adversely affects the organism [30]. Furthermore, an increased level of IgM and decreased level of GLB in MP_9 was observed. Serum glucose is an essential indicator of the host metabolic conditions [31]. IgM facilitates the improvement of immune function in finishing pigs. The linearly reduction in GLU levels by MP was probably associated with the MP active substance 1-deoxynojirimycin, which is a potent α-glucosidase inhibitor [32]. Similarly, for hormones, such as GS and IGF, MP_9 was superior to MP_6 and MP_12, but there was no significant difference compared to the Ctrl, except for ACTH, which may be related to the action of beneficial microbiota in the hindgut. Collectively, diets supplemented with 9% MP had a beneficial effect on the growth of finishing pigs, which may be associated with the function of the active substances in MP, such as mulberry flavonoids and mulberry polysaccharides, and the action of beneficial microbiota, such as *Bifidobacterium* and *Lactobacillus*, but 12% MP may have the opposite effect. It is currently unclear why HDL levels are elevated in MP_12, which needs further investigation.

Antioxidation is one of the primary functions of MP; therefore, we detected the foremost antioxidant enzymes in serum and liver, as well as mRNA expression, pertaining to antioxidant function in muscle and liver to verify its antioxidant capacity, respectively. T-AOC indicates the total antioxidant capacity of the body, and MDA reveals the degree of tissue lipid peroxidation [33]. In the current study, diets supplemented with MP at 9% could increase the levels of T-AOC and decrease the content of MDA, compared with MP_6, but there was no difference compared with the Ctrl. For mRNA expression of liver and muscle, there was no adverse effect between treatments and Ctrl for finishing pigs, indicating MP_9 is more powerful than MP_6 and MP_12, regarding antioxidant function. Our findings are comparable to previous results. Liu et al. [34] reported that diets supplemented with MP at 3%, 6% and 9% in finishing pigs could not impact the antioxidant-related gene expression of muscle with the exception of the MP at 12%. Fan et al. [35] demonstrated that the antioxidant function in serum and muscle with 5% mulberry leaf powder in the diet of finishing pigs was comparable to the control group. Hence, diets supplemented with 9% MP in finishing pigs might be preferable.

The integrity of the intestinal morphology is crucial for the maintenance of normal intestinal function [36]. Shrunken intestinal villi or increased crypt depth, as well as the reduced activity of digestive enzymes, indicate a reduction in nutrient absorption in the small intestine, which are detrimental to nutrient digestibility in pigs [37]. In the current study, diets supplemented with 9% MP had no negative effect on the digestibility of DM, CP and GE, while 6% and 12% had an adverse effect. Likewise, diets supplemented with MP neither affected the activity of digestive enzymes and intestinal morphology nor harmed the intestinal inflammation-related factors, suggesting that the digestive and absorption capability of finishing pigs may not be compromised by the 9% MP. Notably, diets supplemented with MP did not involve microbial diversity (α and β diversity) and composition of the hindgut, only some microbiota will be affected. In the current study, the relative abundance of beneficial microbiota *Bifidobacterium* and *Lactobacillus* were enriched in MP_9, while MP_6 and MP_12 experienced the contrary, which may well be instrumental in explaining the up-regulated mRNA expression of tight junction protein occludin and Mucin-2. Moreover, *Campylobacter* is capable of triggering acute gastroenteritis [38], which was enriched in the Ctrl but not in the MP group, indicating that some active ingredients in MP are able to suppress the proliferation and growth of Campylobacter, which is beneficial to the healthy status of finishing pigs.

*Senegalimassilia* is a specialized microorganism that degrades cellulose [39], which was enriched in MP_12, revealing that high cellulose content of 12% MP led to an elevated abundance of cellulose-degrading microorganisms in the intestine. This also contributes to explaining the reduction in nutrient digestibility and performance in finishing pigs supplemented with 12% MP.

It is well-known that the excessive subcutaneous adipose tissue of fattening pigs is unfavorable for carcass traits [24]. Our findings revealed that the dietary supplementation of MP with increasing levels showed no effect on the carcass trait of finishing pigs, which is in disagreement with the findings of Liu et al. [24] who evidenced that diets supplemented with 9% MP could reduce the back fat thickness and increase the loin-eye area, improving the carcass characteristics. Zhu et al. [25] also revealed that dietary supplementation of 4% MP could ameliorate the carcass trait. Astonishingly, Chen et al. [40] were concordant with our results that the dietary supplementation of MP exhibited no significant effect on carcass qualities of fattening pigs, and the inconsistency in the results may be attributed to differences in pig species, age and composition of MP, which remains to be further verified.

The meat color L* value (luminosity) represents the level of oxidized myoglobin, the a* value (redness) reflects the concentration and presence of deoxygenated myoglobin in the muscle, and the b* value (yellowness) describes the amount of myoglobin oxidized to high-iron myoglobin [41]. Therefore, it is necessary for the meat to have more redness. In the current study, diets supplemented with 9% MP was capable of elevating the a* _24h_, which was helpful for improving the meat quality. Liu et al. [24] reported that diets supplemented with 6% and 9% MP could modulate the shear force, modifying the tenderness and resulting in ameliorating the meat quality. The same results were observed in the study of Chen et al. [40] and Zhang et al. [42]. Additionally, they assumed that the improvement in meat quality might be involved with quercetin, one of the active substances in mulberry leaf powder, which has the potent water-holding ability and antioxidant properties. Interestingly, there were no differences observed between the treatments and the Ctrl in either muscle lipid profile or amino acid composition, suggesting that diets supplemented with MP do not affect the muscle composition of finishing pigs, which also remained in accordance with the findings that there was no significant difference in the mRNA expression levels of genes related to muscle fiber and lipid metabolism between the MP group and the Ctrl. Information concerning the effect of MP on muscle composition in finishing pigs is limited. Hence, it is necessary to deeply investigate the relationship between the active substance related to MP and meat composition.

## 5. Conclusions

In summary of our study, dietary supplementation of 9% MP had no negative effects on growth performance, antioxidative capacity, intestinal digestive enzyme activities and inflammatory factors, carcass traits, the lipid profile and amino acid of muscle, or the related mRNA expression. Furthermore, MP_9 could improve the serum immunity (IgM), enhance the meat quality (a* _24h_), up-regulate the mRNA expression related to the mechanical (occludin) and chemical (Mucin-2) barriers and enrich the beneficial microbiota of hindgut (*Bifidobacterium*). Hence, dietary supplementation of 9% MP in finishing pigs may be advisable.

## Figures and Tables

**Figure 1 antioxidants-11-02243-f001:**
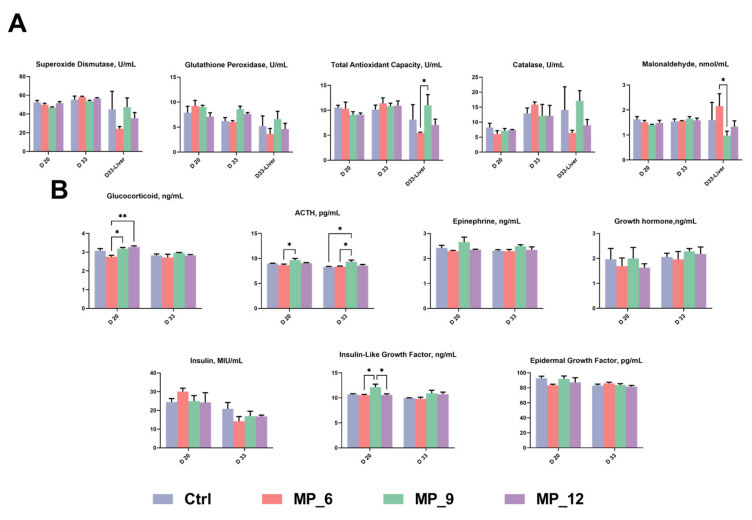
Antioxidant characteristics (**A**) and serum hormone contents (**B**) in finishing pigs dietary supplementation of mulberry leaf powder of varying levels. ACTH, adrenocorticotropic hormone. Ctrl, basal diet with 0% MP; MP_6, MP_9, MP_12, the basal diet supplemented with 6%, 9% and 12% MP, respectively. Data were displayed as mean ± SEM. Bar with different asterisk (*) indicated the degree of significant difference. *n* = 3. *, *p* < 0.05. **, *p* < 0.01.

**Figure 2 antioxidants-11-02243-f002:**
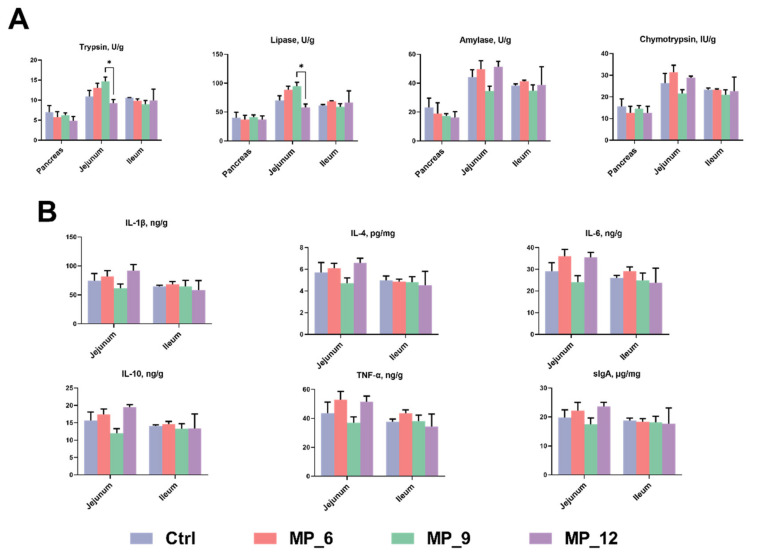
Digestive enzyme activity (**A**) and intestinal inflammatory factors (**B**) in finishing pigs dietary supplementation of mulberry leaf powder of varying levels. IL-1β, interleukin-1β; IL-4, interleukin-4; IL-6, interleukin-6; IL-10 interleukin-10; TNF-α, tumor necrosis factor-α; sIgA, secretory immunoglobulin A. Ctrl, basal diet with 0% MP; MP_6, MP_9 and MP_12 are the batabksal diets supplemented with 6%, 9% and 12% MP, respectively. Data were displayed as mean ± SEM. Bar with different asterisk (*) indicated the degree of significant difference. *n* = 3. *, *p* < 0.05.

**Figure 3 antioxidants-11-02243-f003:**
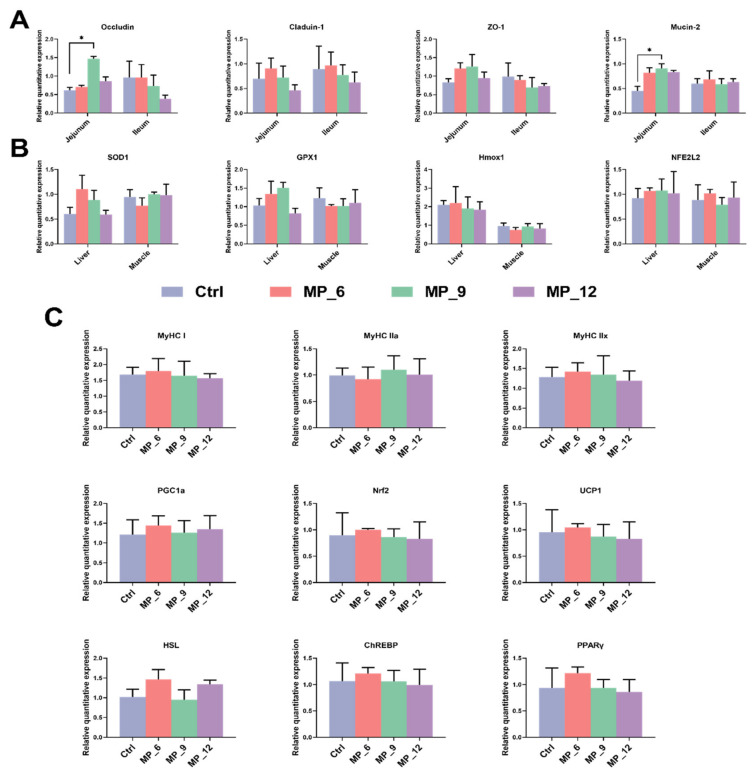
Relative quantitative expression associated with intestinal function (**A**) and antioxidant capacity (**B**), as well as muscle fiber and lipid metabolism (**C**) in finishing pigs dietary supplementation of mulberry leaf powder of varying levels. ZO-1, zonula occludens-1; SOD1, superoxide dismutase; GPX1, glutathione peroxidase 1; Hmox1, heme oxygenase 1; NFE2L2, nuclear factor erythroid 2-like 2; MyHC1, myosin heavy chain 1; MyHC IIa, myosin heavy chain IIa; MyHC IIx myosin heavy chain IIx; PGC-1α, peroxisome proliferator-activated receptor γ coactivator-1α; Nrf2, nuclear respiratory factor; UCP1, uncoupling protein-1; HSL, hormone-sensitive lipase; ChREBP, carbohydrate response element binding protein; PPARγ, peroxisome proliferator-activated receptor γ. Ctrl, basal diet with 0% MP; MP_6, MP_9 and MP_12 are the basal diets supplemented with 6%, 9% and 12% MP, respectively. Data were displayed as mean ± SEM. Bar with different asterisk (*) indicated the degree of significant difference. *n* = 3. *, *p* < 0.05.

**Figure 4 antioxidants-11-02243-f004:**
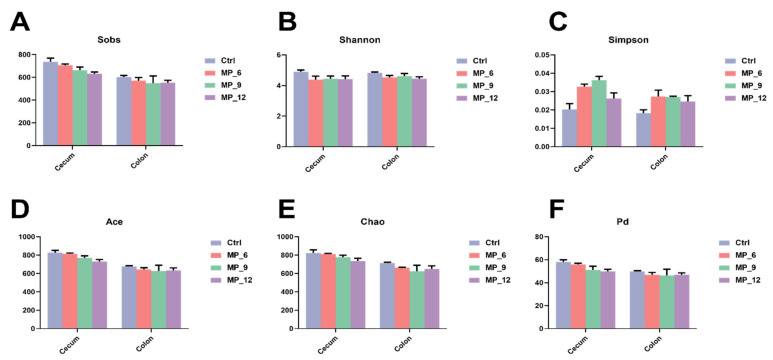
Hindgut microbial α-diversity and rarefaction curves at OTU level in finishing pigs dietary supplementation of mulberry leaf powder of varying levels. (**A**) Sobs; (**B**) Shannon; (**C**) Simpson; (**D**) Ace; (**E**) Chao; (**F**) Phylogenetic diversity. Ctrl, basal diet with 0% MP; MP_6, MP_9 and MP_12 are the basal diets supplemented with 6%, 9% and 12% MP, respectively. Data were displayed as mean ± SEM. *n* = 3.

**Figure 5 antioxidants-11-02243-f005:**
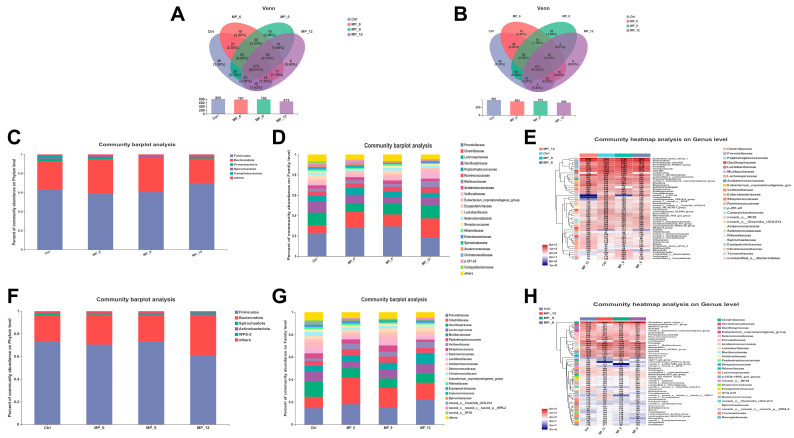
Overview of hindgut microbial composition in finishing pigs dietary supplementation of mulberry leaf powder of varying levels. (**A**) Venn diagram at OTU level in cecum. (**B**) Venn diagram at OTU level in colon. (**C**–**E**) Microbial composition at phylum, family and genus levels in cecum. (**F**–**H**) Microbial composition at phylum, family and genus levels in colon. Ctrl, basal diet with 0% MP; MP_6, MP_9 and MP_12 are the basal diets supplemented with 6%, 9% and 12% MP, respectively. *n* = 3.

**Figure 6 antioxidants-11-02243-f006:**
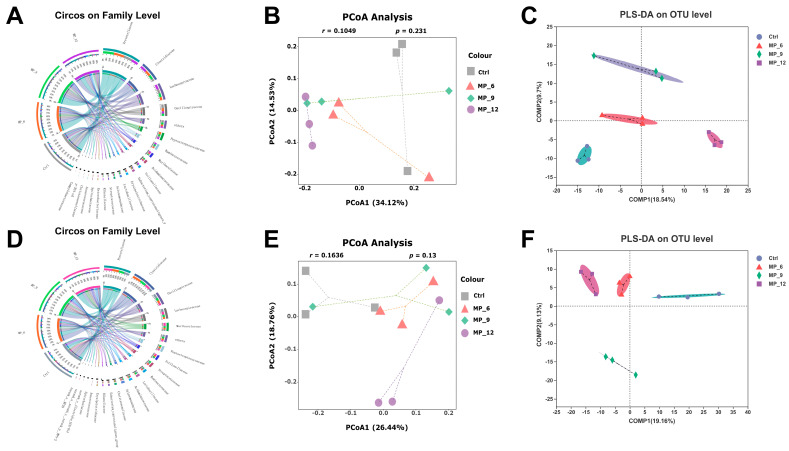
Hindgut microbial β-diversity in finishing pigs dietary supplementation of mulberry leaf powder of varying levels. (**A**) Circos diagram at family level in cecum. (**B**) PCoA at OTU level in cecum. (**C**) PLS-DA at OTU level in cecum. (**D**) Circos diagram at family level in colon. (**E**) PCoA at OTU level in colon. (**F**) PLS-DA at OTU level in colon. In the Circos sample–species relationship diagram, the small half-circle (left half-circle) illustrated the species composition in a sample, where the outer color represented the different groups, the inner color indicated the species and the length denoted the relative abundance of that species in the corresponding sample; the large half-circle (right half-circle) illustrated the proportional distribution of species in different samples at that family level, where the outer color represented the species, the inner color indicated the different groups, and the length denoted the proportional distribution of that sample in a particular species. PCoA, principal co-ordinate analysis, which was evaluated based on the bray_curtis algorithm and ANOSIM difference examination; PLS-DA, partial least squares discriminant analysis. Ctrl, basal diet with 0% MP; MP_6, MP_9 and MP_12 are the basal diets supplemented with 6%, 9% and 12% MP, respectively. *n* = 3.

**Figure 7 antioxidants-11-02243-f007:**
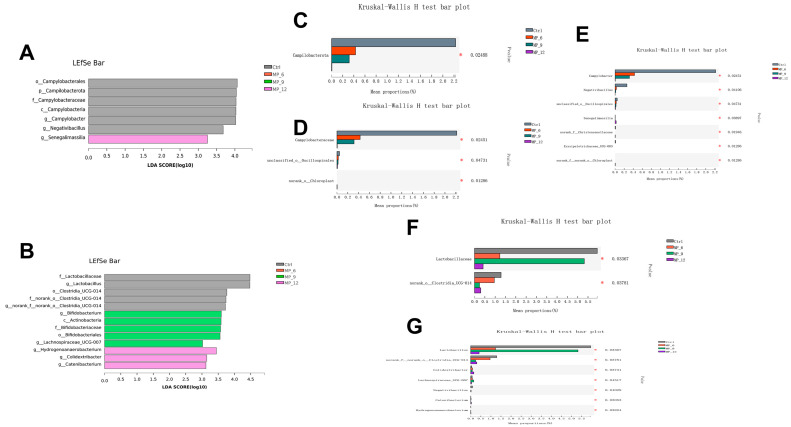
LEfSe analysis from phylum to genus level and histogram of hindgut microbial composition differences in finishing pigs dietary supplementation of mulberry leaf powder of varying levels. (**A**) LDA in cecum. (**B**) LDA in colon. (**C**–**E**) Differences in cecum microbiota at phylum, family and genus levels. (**F**,**G**) Differences in colon microbiota at family and genus levels. The data were analyzed using the Welch’s t-test with FDR correction. LEfSe, linear discriminant analysis effect size; LDA, linear discriminant analysis. *p* < 0.05 and LDA score >3 were displayed. Ctrl, basal diet with 0% MP; MP_6, MP_9 and MP_12 are the basal diets supplemented with 6%, 9% and 12% MP, respectively. Bar with different asterisk (*) indicated the degree of significant difference. *n* = 3. *, *p* < 0.05.

**Table 1 antioxidants-11-02243-t001:** Effect of mulberry leaf powder of varying levels on growth performance in finishing pigs.

Item	Mulberry Leaf Powder				SEM	*p*-Value		
	0	6%	9%	12%		Treatment	Linear	Quadratic
D 1 to 20								
ADG, kg/d	0.71	0.67	0.67	0.64	0.02	0.16	0.04	0.95
ADFI, kg/d	2.66	2.62	2.61	2.51	0.08	0.61	0.27	0.60
FCR	3.79	3.92	3.92	3.94	0.10	0.68	0.29	0.66
D 21 to 33								
ADG, kg/d	0.77	0.66	0.71	0.68	0.02	0.06	0.05	0.13
ADFI, kg/d	2.87	2.78	2.80	2.83	0.02	0.05	0.09	0.02
FCR	3.75	4.20	3.94	4.15	0.13	0.14	0.10	0.37
D 1 to 33								
ADG, kg/d	0.74 ^a^	0.67 ^b^	0.69 ^ab^	0.66 ^b^	0.02	0.04	0.01	0.38
ADFI, kg/d	2.77	2.70	2.71	2.67	0.05	0.63	0.24	0.92
FCR	3.77	4.06	3.93	4.05	0.11	0.29	0.14	0.47

SEM, standard error of the mean; ADG, average daily gain; ADFI, average daily feed intake; FCR, feed conversion ratios. ^a,b^ Within a row, values with different superscripts are significantly different (*p* < 0.05). *n* = 3.

**Table 2 antioxidants-11-02243-t002:** Effect of mulberry leaf powder of varying levels on serum immunity and inflammatory factors in finishing pigs.

Item	Mulberry Leaf Powder				SEM	*p*-Value		
	0	6%	9%	12%		Treatment	Linear	Quadratic
D 22								
IgA, μg/mL	15.06	14.33	13.90	15.15	0.78	0.65	0.83	0.30
IgG, mg/mL	7.00	6.76	6.42	7.72	0.48	0.35	0.56	0.17
IgM, μg/mL	6.47	6.00	6.19	6.64	0.17	0.13	0.69	0.03
IL-1β, ng/L	102	100	110	110	3.61	0.18	0.10	0.34
IL-6, ng/L	34.75	32.67	33.53	35.05	1.29	0.57	0.97	0.19
IL-8, ng/L	72.27	63.65	65.11	72.98	2.11	0.04	0.76	0.01
IL-10, ng/L	22.08	19.87	19.58	20.64	0.64	0.12	0.09	0.07
TNF-α, ng/L	51.74	49.48	48.73	52.80	2.42	0.63	0.98	0.26
IFN-γ, pg/mL	144 ^b^	145 ^b^	159 ^b^	179 ^a^	5.31	0.01	0.00	0.03
D 33								
IgA, μg/mL	13.62	14.19	14.33	13.08	0.47	0.31	0.71	0.10
IgG, mg/mL	6.74	7.24	6.85	6.77	0.33	0.71	0.98	0.35
IgM, μg/mL	5.94 ^b^	5.70 ^b^	6.44 ^a^	6.11 ^ab^	0.11	0.02	0.06	0.45
IL-1β, ng/L	104	106	107	101	1.44	0.10	0.60	0.03
IL-6, ng/L	33.91	33.41	33.94	32.91	0.98	0.86	0.60	0.80
IL-8, ng/L	64.21	65.29	66.00	63.60	1.68	0.75	1.00	0.37
IL-10, ng/L	18.54	19.75	18.99	18.21	0.56	0.33	0.76	0.10
TNF-α, ng/L	49.07	49.11	51.92	51.64	2.04	0.65	0.31	0.81
IFN-γ, pg/mL	151	143	171	160	10.03	0.34	0.33	0.74

SEM, standard error of the mean; IgA, G, M, immunoglobulin A, G, M; IL-1β, 6, 8, 10, interleukin, 6, 8, 10; TNF-α, tumor necrosis factor-α; IFN-γ, interferon-γ. ^a,b^ Within a row, values with different superscripts are significantly different (*p* < 0.05). *n* = 3.

**Table 3 antioxidants-11-02243-t003:** Effect of mulberry leaf powder of varying levels on serum metabolite in finishing pigs.

Item	Mulberry Leaf Powder				SEM	*p*-Value		
	0	6%	9%	12%		Treatment	Linear	Quadratic
D 22								
GLU, mmol/L	7.20 ^a^	6.24 ^b^	5.80 ^b^	5.45 ^b^	0.20	0.01	0.03	0.06
TC, mmol/L	2.34	2.18	2.41	2.81	0.21	0.29	0.20	0.16
TG, mmol/L	0.51	0.57	0.73	0.54	0.07	0.23	0.37	0.23
HDL, mmol/L	0.73 ^b^	0.86 ^ab^	0.82 ^ab^	0.99 ^a^	0.05	0.04	0.01	0.52
LDL, mmol/L	1.58	1.53	1.66	2.02	0.13	0.14	0.08	0.11
ALT, U/L	55.32	36.16	40.31	34.75	4.51	0.06	0.02	0.26
AST, U/L	88.50	43.13	37.68	51.63	21.59	0.41	0.20	0.29
TP, g/L	67.31	65.03	68.92	69.20	4.81	0.92	0.73	0.69
ALB, g/L	34.80	31.67	33.13	30.82	2.54	0.71	0.36	0.91
GLB, g/L	32.39	31.43	33.35	39.91	2.27	0.13	0.08	0.09
ALP, U/L	134.32 ^a^	128.88 ^a^	133.74 ^a^	78.23 ^b^	5.21	0.00	0.00	0.00
LDH, U/L	821	515	630	521	87.42	0.14	0.06	0.34
UA, μmol/L	21.12	29.05	34.30	20.70	3.72	0.11	0.54	0.04
BUN, mmol/L	2.25	2.55	1.24	2.39	0.62	0.49	0.76	0.74
DLA, μmol/L	7.17	6.85	6.84	7.04	0.15	0.41	0.41	0.16
NEFA, μmol/L	178	158	167	198	8.94	0.08	0.27	0.02
D 33								
GLU, mmol/L	10.52	9.37	9.30	8.16	3.60	0.78	0.87	0.69
TC, mmol/L	2.92	2.36	2.87	2.65	0.24	0.40	0.61	0.37
TG, mmol/L	1.32	1.10	1.24	1.29	0.21	0.88	0.92	0.50
HDL, mmol/L	0.84	0.73	1.03	0.78	0.09	0.22	0.82	0.78
LDL, mmol/L	1.99	1.62	1.96	1.94	0.14	0.31	0.96	0.17
ALT, U/L	56.76	57.19	64.92	61.42	4.48	0.57	0.33	0.95
AST, U/L	194	196	141	182	58.24	0.90	0.73	0.89
TP, g/L	76.34	81.25	74.90	76.16	3.18	0.55	0.79	0.41
ALB, g/L	39.51	33.54	42.17	38.82	1.91	0.09	0.70	0.22
GLB, g/L	36.84 ^ab^	44.48 ^a^	32.67 ^b^	35.68 ^ab^	2.04	0.03	0.29	0.08
ALP, U/L	133	126	142	111	23.90	0.83	0.68	0.67
LDH, U/L	764	638	762	793	118	0.79	0.82	0.43
UA, μmol/L	46.95	37.84	39.58	38.95	5.25	0.63	0.31	0.50
BUN, mmol/L	3.42	3.66	4.04	4.26	0.22	0.13	0.03	0.59
DLA, μmol/L	6.41	6.50	6.60	6.14	0.13	0.19	0.41	0.09
NEFA, μmol/L	173	169	159	161	2.97	0.05	0.02	0.82

SEM, standard error of the mean; GLU, glucose; TC, total cholesterol; TG, total triglycerides; HDL, high-density lipoprotein; LDL, low-density lipoprotein; ALT, alanine aminotransferase; AST, aspartate aminotransferase; TP, total protein; ALB, albumin; GLB, globulin; ALP, alkaline phosphatase; LDH, lactate dehydrogenase; UA, uric acid, BUN, blood urea nitrogen; DLA, D-lactate. ^a,b^ Within a row, values with different superscripts are significantly different (*p* < 0.05). *n* = 3.

**Table 4 antioxidants-11-02243-t004:** Effect of mulberry leaf powder of varying levels on nutrient digestibility in finishing pigs.

Item	Mulberry Leaf Powder				SEM	*p*-Value		
	0	6%	9%	12%		Treatment	Linear	Quadratic
Dry matter	0.79 ^a^	0.67 ^b^	0.79 ^a^	0.69 ^b^	0.02	0.01	0.01	0.09
Organic matter	0.83	0.82	0.73	0.75	0.03	0.07	0.14	0.43
Crude protein	0.70 ^a^	0.51 ^b^	0.70 ^a^	0.55 ^b^	0.02	0.01	0.01	0.11
Gross energy	0.78 ^a^	0.64 ^b^	0.76 ^a^	0.67 ^b^	0.01	0.01	0.01	0.05
Neutral detergent fiber	0.52	0.53	0.56	0.55	0.02	0.71	0.34	0.30
Acid detergent fiber	0.38	0.30	0.30	0.30	0.03	0.63	0.24	0.92
Ether extract	0.51	0.53	0.39	0.37	0.05	0.09	0.08	0.86
Calcium	0.31	0.33	0.22	0.27	0.04	0.34	0.27	0.87
Phosphorus	0.13	0.22	0.15	0.13	0.04	0.06	0.80	0.02

SEM, standard error of the mean. ^a,b^ Within a row, values with different superscripts are significantly different (*p* < 0.05). *n* = 3.

**Table 5 antioxidants-11-02243-t005:** Effect of mulberry leaf powder of varying levels on carcass characteristics and meat quality in finishing pigs.

Item	Mulberry Leaf Powder				SEM	*p*-Value		
	0	6%	9%	12%		Treatment	Linear	Quadratic
Carcass characteristics								
Carcass weight, kg	69.73	68.27	66.63	66.30	0.98	0.14	0.03	0.99
Carcass length, cm	78.23	77.63	74.80	75.97	0.89	0.11	0.05	0.82
Shoulder fat thickness, mm	26.50	31.89	32.77	31.15	2.03	0.23	0.10	0.21
Lumbosacral fat thickness, mm	13.44	21.39	13.44	13.97	3.31	0.33	0.92	0.19
The 6th to 7th rib fat thickness, mm	19.59	23.91	20.48	23.21	2.52	0.59	0.46	0.70
The 10th rib fat thickness, mm	17.93	21.56	18.76	21.56	2.08	0.53	0.36	0.79
The last rib fat thickness, mm	20.34	16.83	21.61	20.56	2.96	0.69	0.81	0.53
Loin eye muscle area, cm^2^	37.09	32.36	37.84	34.78	4.10	0.78	0.86	0.73
Dressing percentage, %	68.61	68.25	68.49	66.88	0.88	0.52	0.29	0.45
Meat quality								
Flesh color score	3.37	3.33	3.10	3.17	0.19	0.71	0.36	0.98
L*_45min_	43.64	43.26	44.73	44.67	2.14	0.95	0.68	0.83
a*_45min_	4.58	5.06	5.87	4.31	0.62	0.38	0.85	0.21
b*_45min_	2.11	2.82	2.75	3.18	0.65	0.72	0.30	0.93
pH_45min_	6.00	6.12	5.85	5.95	0.14	0.59	0.58	0.67
L*_24h_	51.81	50.77	51.6	53.23	1.76	0.80	0.64	0.42
a*_24h_	6.83 ^ab^	7.16 ^ab^	8.20 ^a^	5.49 ^b^	0.72	0.05	0.48	0.06
b*_24h_	6.79	6.72	7.10	5.78	0.72	0.62	0.51	0.41
pH_24h_	5.50	5.62	5.53	5.62	0.05	0.22	0.16	0.75
Drip loss, %	1.50	1.02	1.09	1.07	0.30	0.66	0.32	0.53
Marbling score	2.50	2.67	2.47	2.53	0.12	0.66	0.98	0.53

SEM, standard error of the mean; L*_45min_, a*_45min_, b*_45min_, luminosity, redness and yellowness at 45 min; L*_24h_, a*_24h_, b*_24 h,_ luminosity, redness and yellowness at 24 h. ^a,b^ Within a row, values with different superscripts are significantly different (*p* < 0.05). *n* = 3.

**Table 6 antioxidants-11-02243-t006:** Effect of mulberry leaf powder of varying levels on concentration of muscle amino acid in finishing pigs (mg/g).

Item	Mulberry Leaf Powder				SEM	*p*-Value		
	0	6%	9%	12%		Treatment	Linear	Quadratic
Aspartic acid	7.45	7.01	7.35	7.56	0.19	0.30	0.68	0.10
Glutamic acid	11.89	11.15	11.76	12.07	0.30	0.26	0.68	0.09
Threonine	3.61	3.40	3.57	3.67	0.09	0.31	0.66	0.10
Serine	3.02	2.83	2.96	3.05	0.07	0.28	0.78	0.08
Proline	2.89	2.73	2.83	2.94	0.05	0.12	0.63	0.03
Glycine	3.45	3.28	3.38	3.48	0.08	0.41	0.84	0.12
Alanine	4.51	4.25	4.43	4.55	0.11	0.30	0.80	0.09
Cysteine	0.91	0.88	0.91	0.93	0.02	0.31	0.51	0.12
Valine	4.24	4.02	4.19	4.31	0.11	0.35	0.71	0.11
Methionine	2.27	2.27	2.32	2.37	0.05	0.48	0.19	0.45
Isoleucine	3.99	3.76	3.94	4.06	0.11	0.34	0.67	0.12
Leucine	6.54	6.18	6.45	6.63	0.17	0.37	0.73	0.12
Tyrosine	2.75	2.59	2.73	2.78	0.07	0.27	0.72	0.10
Phenylalanine	3.23	3.04	3.19	3.26	0.08	0.26	0.78	0.08
Histidine	4.05	3.69	3.77	3.90	0.10	0.16	0.25	0.06
Lysine	7.28	6.89	7.21	7.42	0.19	0.34	0.62	0.12
Arginine	5.49	5.11	5.52	5.56	0.11	0.07	0.48	0.04
Tryptophan	1.05	0.99	1.03	1.06	0.03	0.44	0.83	0.14

SEM, standard error of the mean; *n* = 3.

**Table 7 antioxidants-11-02243-t007:** Effect of mulberry leaf powder of varying levels on lipid profile in finishing pigs.

Item	Mulberry Leaf Powder				SEM	*p*-Value		
	0	6%	9%	12%		Treatment	Linear	Quadratic
C10:0	0.09	0.11	0.10	0.09	0.01	0.19	0.52	0.06
C12:0	0.14	0.11	0.10	0.14	0.02	0.31	0.66	0.11
C14:0	1.14	1.27	1.11	1.25	0.12	0.71	0.70	0.90
C15:0	0.06	0.04	0.06	0.05	0.01	0.39	0.93	0.61
C16:0	23.91	24.83	24.31	25.24	0.74	0.63	0.32	0.97
C16:1	2.85	2.67	2.65	2.68	0.13	0.70	0.34	0.55
C17:0	0.29	0.24	0.32	0.28	0.03	0.26	0.80	0.53
C18:0	13.40	14.76	13.85	14.36	0.49	0.32	0.30	0.37
C18:1n9c	38.68	40.70	37.33	38.73	1.45	0.49	0.75	0.58
C18:2n6c	13.08	10.58	13.25	11.42	1.19	0.39	0.56	0.62
C18:3n3	0.34	0.38	0.36	0.35	0.03	0.81	0.69	0.45
C20:0	0.41	0.31	0.41	0.41	0.05	0.43	0.83	0.23
C20:1	0.71	0.76	0.75	0.81	0.05	0.66	0.29	0.85
C21:0	0.54	0.50	0.57	0.51	0.05	0.75	0.95	0.99
C20:2	0.13	0.08	0.16	0.12	0.03	0.48	0.81	0.66
C20:3n6	0.37	0.27	0.43	0.35	0.08	0.54	0.88	0.63
C20:4n6	3.05	1.84	3.39	2.46	0.79	0.56	0.85	0.69
C20:3n3	0.08	0.08	0.08	0.09	0.01	0.92	0.65	0.66
C22:0	0.08	0.05	0.10	0.06	0.02	0.35	0.96	0.95
C20:5n3	0.11	0.07	0.12	0.10	0.03	0.74	0.93	0.71
C22:1n9	0.06	0.03	0.05	0.04	0.01	0.29	0.24	0.57
C22:2	0.07	0.06	0.07	0.07	0.01	0.94	0.81	0.71
C24:0	0.08	0.05	0.08	0.08	0.02	0.68	0.80	0.41
C24:1	0.13	0.07	0.12	0.11	0.02	0.37	0.70	0.22
C22:6n3	0.21	0.13	0.21	0.17	0.06	0.75	0.81	0.62
SFA	40.13	42.28	41.01	42.49	1.16	0.49	0.26	0.78
MUFA	42.64	44.39	41.15	42.57	1.41	0.50	0.70	0.64
Total N-3 FA	0.73	0.66	0.77	0.71	0.09	0.83	0.95	0.82
Total N-6 FA	16.50	12.69	17.07	14.23	2.00	0.44	0.68	0.64
PUFA	17.23	13.34	17.84	14.94	2.08	0.45	0.69	0.64
UFA	59.87	57.72	58.99	57.51	1.16	0.49	0.26	0.78
N-6/N-3	22.46	19.38	22.14	20.10	0.93	0.14	0.24	0.45
P/S	0.43	0.32	0.44	0.35	0.06	0.47	0.60	0.70

SEM, standard error of the mean; SFA: saturated fatty acids; MUFA: monounsaturated fatty acids; Total N-3 FA; Total N-3 fatty acids; Total N-6 FA: Total N-6 fatty acids; PUFA: polyunsaturated fatty acids; UFA: unsaturated fatty acids (MUFA + PUFA); N-6/N-3: Total N-6 FA / Total N-3 FA; P/S: PUFA/SFA. *n* = 3.

**Table 8 antioxidants-11-02243-t008:** Effect of mulberry leaf powder of varying levels on intestinal morphology in finishing pigs.

Item	Mulberry Leaf Powder				SEM	*p*-Value		
	0	2%	4%	6%		Treatment	Linear	Quadratic
Duodenum								
Villus height	335	288	225	197	51.11	0.31	0.08	0.82
Crypt depth	430	478	418	430	41.43	0.75	0.86	0.53
VH/CD	0.76	0.64	0.56	0.46	0.12	0.42	0.12	0.84
Jejunum								
Villus height	274	239	225	182	29.30	0.27	0.07	0.66
Crypt depth	346	317	319	348	33.49	0.86	0.94	0.42
VH/CD	0.79	0.76	0.73	0.56	0.12	0.54	0.24	0.47
Ileum								
Villus height	298	305	240	218	43.57	0.47	0.20	0.51
Crypt depth	325	355	288	277	40.94	0.56	0.36	0.43
VH/CD	0.92	0.90	0.86	0.83	0.16	0.98	0.70	0.91

SEM, standard error of the mean; VH/CD, the ratio of villus height to crypt depth. *n* = 3.

## Data Availability

The original contributions generated for this study are included in the article; further inquiries can be directed to the corresponding author.

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
