# Peer review of "Effect of Mulberry Leaf Powder of Varying Levels on Growth Performance, Immuno-Antioxidant Status, Meat Quality and Intestinal Health in Finishing Pigs"

_antioxidants, 2022, doi:10.3390/antiox11112243_

Round 1
Reviewer 1 Report
The current ms is well-written, however, I list some mis-spelling and writing suggestions as follows,
Line 21 inflammatory
Line 26 Bacteria Genus, Italic
Line 29 inflammatory
Line 100 ad libitum, Italic
I suggest ignoring Tables 6, 7,9, and Fig 3 because there were no significant results and moving to the appendix Table and Figure.
In Table S1, Gross energy is wrong spelling
Conclusion and Abstract, please amend the wording “: In a nutshell, in our study,” into “Summarized our study, ” which will be briefer.
The experimental basal diet was formulated based on iso-caloric and iso-protein. Hence, please make some possible clear explanations of current results that MP6 and MP12 show a detrimental effect on performance and digestibility. However, MP9 did not have these findings, data from table 1, D1-D33 showed linear P value<0.01, but no quadratic was found.
Author Response
The current ms is well-written, however, I list some mis-spelling and writing suggestions as follows:
Many thanks to the reviewers for your positive feedback on our manuscript and also for their very helpful and constructive comments to better improve the quality of the manuscript. All modifications have been highlighted in red color. We would like to express our most sincere appreciation for your seriousness and efforts.
Line 21 inflammatory
Many thanks for your comments, we apologized for our negligence, we have revised and checked the similar issue in our manuscript.
Line 26 Bacteria Genus, Italic
Many thanks for your comments, we apologized for our negligence, we have revised and checked the similar issue in our manuscript.
Line 29 inflammatory
Many thanks again for your comments, we apologized for our negligence, we have revised and checked the similar issue in our manuscript.
Line 100 ad libitum, Italic
Many thanks for your comments, we have revised in our manuscript.
I suggest ignoring Tables 6, 7, 9 and Fig 3 because there were no significant results and moving to the appendix Table and Figure.
Many thanks for your comments, based on your and other reviewers' suggestions, we decided to put Figure 3 into supplementary material, because amino acids and fatty acids are relatively important parts for meat quality, and there was no significant difference when raw wheat bran was replaced with mulberry powder, which was a result of positive feedback.
In Table S1, Gross energy is wrong spelling
Many thanks for your comments, we apologized for our negligence, we have revised in our manuscript.
Conclusion and Abstract, please amend the wording “: In a nutshell, in our study,” into “Summarized our study, ” which will be briefer.
Many thanks for your comments, we have revised in our manuscript according to your suggestion.
The experimental basal diet was formulated based on iso-caloric and iso-protein. Hence, please make some possible clear explanations of current results that MP6 and MP12 show a detrimental effect on performance and digestibility. However, MP9 did not have these findings, data from table 1, D1-D33 showed linear P value<0.01, but no quadratic was found.
Many thanks for your comments, we compared the composition of wheat bran in NRC with the nutrient composition of current mulberry powder, and we found that the content of related essential amino acids (such as arginine, leucine, etc.) in mulberry powder was lower than that of wheat bran. Therefore, we speculate that the decline in the growth performance of fattening pigs caused by 6% mulberry powder may be related to the lower essential amino acids of mulberry powder. With the increase of mulberry powder concentration (9%), some increased active substances in mulberry powder such as flavonoids and polyphenolic compounds has played a positive role in improving immune and intestinal health. Additionally, the reason that MP_12 show a detrimental effect on performance and digestibility may be attributed to the high concentration of fiber in the MP compared with wheat bran. Researcher has revealed that high-fiber diets have a detrimental effect on energy and nutrient absorption, leading to decreased growth performance in finishing pigs. We have added it in our discussion.
Reviewer 2 Report
General comment: The authors reported the effect of dietary supplementation of mulberry leaf powder on growth performance, health biomarkers, and meat quality of finishing pigs. The authors need to better justify the use of mulberry leaf powder as a dietary supplement for a monogastric animal instead of ruminant animals. Although the authors performed a thorough investigation, the data are poorly organized, presented, and discussed. In addition, many grammatical errors and awkward syntax were found. The authors may consider referring the manuscript to a native English speaker to revise it.
Specific comments:
Line 2: The term “mulberry powder” is misleading. It sounds like powder made of mulberry. But the authors used powder made of predominantly mulberry leaves. It should be changed to “mulberry leaf powder” throughout the manuscript.
Lines 2-5: The title is too verbose and has grammatical errors.
Line 13: “mulberry”.
Line 18: “MP” needs to be defined first.
Lines 23-24: Define all the abbreviations.
Line 40: Give a few examples of the unconventional feed ingredients.
Line 282: Which post hoc test was used?
Lines 341-342: It seems MP_9 only had a higher a* than MP_12. Please double check.
Figures 5G and 5H should be supplementary information.
Lines 432-437: Redundant paragraph.
Figures 6 and 7 are too crowded to be legible.
Author Response
General comment: The authors reported the effect of dietary supplementation of mulberry leaf powder on growth performance, health biomarkers, and meat quality of finishing pigs. The authors need to better justify the use of mulberry leaf powder as a dietary supplement for a monogastric animal instead of ruminant animals. Although the authors performed a thorough investigation, the data are poorly organized, presented, and discussed. In addition, many grammatical errors and awkward syntax were found. The authors may consider referring the manuscript to a native English speaker to revise it.
Many thanks to the reviewers for your positive feedback on our manuscript and also for their very helpful and constructive comments to better improve the quality of the manuscript. All modifications have been highlighted in red color. We would like to express our most sincere appreciation for your seriousness and efforts. Also, we have done a thorough check of the grammatical content of the context, thank you again for your comments and suggestions.
Specific comments:
Line 2: The term “mulberry powder” is misleading. It sounds like powder made of mulberry. But the authors used powder made of predominantly mulberry leaves. It should be changed to “mulberry leaf powder” throughout the manuscript.
Many thanks for your suggestion, we have revised and replaced the mulberry powder as mulberry leaf powder throughout the manuscript
Lines 2-5: The title is too verbose and has grammatical errors.
Many thanks for your comments, based on the main test content, we have revised the title as effect of mulberry leaf powder with varying levels on growth performance, immuno-antioxidant status, meat quality and intestinal health in finishing pigs
Line 13: “mulberry”.
Many thanks for your comments, we apologized for our negligence, we have revised and checked the similar issue in our manuscript.
Line 18: “MP” needs to be defined first.
Many thanks for your comments, we have modified and defined it.
Lines 23-24: Define all the abbreviations.
Many thanks for your comments, we have defined it with the full name, such as ADG (average daily gain), DM (dry matter), GE (gross energy), IgM (immunoglobulin M) et al.
Line 40: Give a few examples of the unconventional feed ingredients.
Many thanks for your comments, There are a wide range of the unconventional feed ingredients such as brown rice, corn germ meal, wheat gluten meal, distillers dried grains with solubles and so on. We have supplemented it in our manuscript.
Line 282: Which post hoc test was used?
Many thanks for your comments, we apologized for our negligence about Statistical Analysis. The one-way ANOVA with generalized linear models procedure and the Turkey-Kramer post-hoc test were performed by SAS 9.2 software. We have revised it.
Lines 341-342: It seems MP_9 only had a higher a* than MP_12. Please double check.
Many thanks for your comments, we apologized for our negligence. We have revised and checked the similar issue in our manuscript.
Figures 5G and 5H should be supplementary information.
Many thanks for your comments, we present some of the information (The rarefaction curves of cecum and colon in finishing pigs) in Figure 5G and 5H as supplementary material according to your suggestions.
Lines 432-437: Redundant paragraph.
Many thanks for your comments, This part mainly explains the content of rarefaction curves, we have removed it based on your suggestion in the last question (supplementary material).
Figures 6 and 7 are too crowded to be legible.
Many thanks for your comments, we apologized for for the blurred information caused by Figure (6 and 7) compression when Word was converted into PDF. However, we guarantee that the Word and Tiff figure we submited in the submission system are recognizable and clear.
Round 2
Reviewer 2 Report
The authors have addressed all my comments and revised the manuscript accordingly. My only suggestion is to delete "with varying levels" in the title or change it to "at varying levels".